

# Emerging crescentic patterns in modeled double sandbar systems

Giovanni Coco[1], Daniel Calvete[2], Francesca Ribas[2], Huib E. de Swart[3], and Albert Falques[2]

[1]School of Environment, Faculty of Science, University of Auckland, Auckland, New Zealand
[2]Dep. Física, Universitat Politècnica de Catalunya, Barcelona, Spain
[3]Institute for Marine and Atmospheric Research Utrecht, Utrecht University, Utrecht, The Netherlands

**Correspondence:** G. Coco (g.coco@auckland.ac.nz)

**Abstract.** The morphodynamic stability of double-barred beaches is explored using a numerical model based on linear stability analysis. Surfzone hydrodynamics is described by coupling depth and wave averaged conservation of mass and momentum with the wave-energy and phase equations, including roller dynamics. A simplified sediment transport formulation relates flow to seabed changes. Under normal wave incidence an alongshore uniform coast with a cross-shore profile characterized by the presence of two sandbars, can be unstable, thereby resulting in the development of crescentic/rip channel patterns. Our study demonstrates that sandbar coupling can be either in-phase (highs and lows of both sandbars are at the same alongshore position) or out-of-phase (highs and lows of one sandbar correspond to lows and highs of the other sandbar). In line with observations, results of numerical simulations show a large variability in the possible emerging bottom patterns. Our analysis indicates that the inner bar-modes are dominant for large height/depth differences between the two sandbars crests and small offshore wave heights, while patterns related to the outer sandbar dominate for small values of the difference in sandbar depth. For intermediate differences between the two sandbars depths, patterns on both longshore bars appear to be fully coupled. For relatively larger waves and large depth over the outer sandbar, patterns develop close to the shoreline/inner surfzone.

## 1 Introduction

Multiple sandbar systems have been observed in a variety of settings worldwide. We specifically focus on the dynamics of double sandbar systems in the surfzone where the sandbars almost constantly affect (and are affected by) wave transformation and onshore/offshore exchanges of sediment. Alongshore changes in double sandbar configurations sometimes result in rhythmic patterns, usually called crescentic bars or rip channels (Figure 1).

The development of alongshore patterns in multiple sandbar settings has been studied through both observations (e.g. Castelle et al., 2007, 2015) and numerical studies (e.g. Klein and Schuttelaars, 2006; Price and Ruessink, 2013), and has also been considered in the wider framework of a conceptual model of sequential beach changes by Short and Aagaard (1993). This conceptual model as well as field data (e.g. Castelle et al., 2007) indicate that the inner and outer sandbars are likely to be characterized by different spatial and temporal scales. In a double-barred system like the one considered in this study, the inner sandbar usually displays crescentic features with an alongshore spacing (distance between consecutive sandbar horns) smaller than the one characterizing the outer sandbar. Moreover, in the case of accretionary conditions, Short and Aagaard (1993) assume that the inner sandbar responds faster than the outer one. The Short and Aagaard (1993) model also indicates that

(c) Author(s) 2019. CC BY 4.0 License.





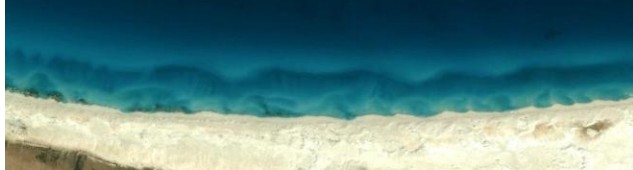

**Figure 1.** Multiple sandbars along the Libian coast (Image from Google, DigitalGlobe 2012).

beach configurations can involve coupling between the sandbars and/or coupling between the inner sandbar and the shoreline. We here use the term "coupling" to indicate the development of crescentic inner/outer sandbar configurations that are either in phase or out of phase. Ruessink et al. (2007) used wavelet analysis to show that inner sandbar alongshore patterns become coupled to the pattern of the outer sandbar. Coupling was concurrent with the onshore migration of the outer sandbar whose

alongshore shape was characterized by the presence of crescents so that when the two sandbars became close, the inner bar developed an alongshore variability in response to the onshore propagating outer bar. Using a 9.3 year dataset of video images collected at low tide on the Gold Coast (Australia), Price and Ruessink (2011, 2013) showed that coupling between the offshore and inner sandbar occurred for 40% of the available observations. Out of the coupled sandbar patterns, the in-phase coupling occurred 85% of the times. Finally, changes in wave height or angle of wave approach can determine both the alongshore

shape of each of the sandbars and control the possible coupling configuration (see also Thiebot et al., 2012). Castelle et al. (2015) describe a variety of coupling patterns occurring along the French coast and used satellite and video imagery to show the occurrence of in phase or out of phase coupled configurations. It should be pointed out that while remote sensing provides increasing evidence of coupling between sandbars, bathymetric surveys providing details about the geometry of the system remain scarce and sparse. More observations are available to describe the coupling between shoreline and sandbar patterns

(e.g., Coco et al., 2005; Ruessink et al., 2007; Price et al., 2014).

The conditions leading to transitions from alongshore uniform to variable have been ascribed to wave height (or wave power) and/or to parameters combining hydro- and sedimentological characteristics (e.g., sediment fall velocity or Iribarren number). More specifically, the development of alongshore variability or the straightening of crescentic sandbars have been ascribed to low- and high-energy events, respectively. Recent observations (Price and Ruessink, 2013) showed instead that

changes in sandbar morphology (from alongshore uniform to variable and vice-versa) do not follow a straightforward cause-effect relationship and that changes in the double sandbar system can be driven by a variety of interconnected factors (from wave angle to preceding bathymetry). Video imagery provides high resolution in time and the large spatial coverage but is not necessarily capable to provide detailed measurements of the geometry of the sandbars (a notable exception is provided by Price et al. (2013)) and the sensitivity to preceding conditions requires more attention.

With respect to the formative mechanism, crescentic sandbars have attracted the attention of nearshore scientists for decades. Initially, their appearance has been ascribed to the presence of a template in the hydrodynamic forcing, edge waves. Edge waves would provide regular alongshore amplitude variations in the hydrodynamics that could be reflected onto the sandbar configu-





ration (Bowen and Inman, 1971; Huntley, 1980; Holman and Bowen, 1982; Aagaard, 1991). A different approach focusing on feedbacks between hydrodynamics, sediment transport and morphological change, indicates that the pattern could emerge as a

result of self-organizing processes (Coco and Murray, 2007). This approach is based on the possibility that nonlinear coupling between hydrodynamics and sediment transport can control and actually promote the evolution of seabed perturbations eventually resulting in a spatially regular (and self-organized) pattern. In terms of both field observations and numerical modelling most of the studies addressing the emergence of crescentic patterns have primarily focused on planar (e.g. Falqués et al., 2000; Caballeria et al., 2002) and single-barred (e.g. Caballeria et al., 2002; Damgaard et al., 2002; Reniers et al., 2004) beaches.

Many studies followed and analyzed various aspects of crescentic sandbar formation: from the influence of settings typical of embayed beaches (e.g. Castelle and Coco, 2012) to the influence of time-varying forcing (Castelle and Ruessink, 2011) and offshore bathymetric perturbations (Castelle et al., 2012).

   In this contribution we aim to systematically address the role of initial bathymetry on the coupling between sandbars, an area that so far has received only limited attention (see also  Price et al., 2014). Calvete et al. (2007) used linear stability

analysis to show that the initial cross-shore beach profile can be as important as wave height in determining the growth rate and alongshore spacing of crescentic bars. Other numerical studies of morphological evolution of double barred beaches also use linear stability analysis to analyze the nonlinear depth- and wave-averaged equations coupled to sediment transport and morphological evolution. The work of Klein and Schuttelaars (2006) for example showed that the magnitude of the longshore current and wave height are directly related to the preferred spacing and the growth rate, respectively. Numerical simulations

of oblique incident waves on double sand bar systems (Klein and Schuttelaars, 2006; Price et al., 2013) show that the coupling between the two sandbars occurs through the development of a meandering alongshore current. In agreement with field observations, numerical simulations (Smit et al., 2008) have also shown that the outer sandbar develops into a crescentic system characterized by a larger spacing than that of the inner sandbar and attributed such difference to the larger water depth of the outer sandbar crest. Relaxing the assumption of depth-averaged motions, and accounting for the circulation currents associated

to undertow still results in the development of a coupled double sandbar system (Dronen and Deigaard, 2007). These studies, although reproducing the emergence of alongshore variability in double sandbar systems do not address the possible coupling between the two sandbars.

   More recently, for the case of normally incident waves, Castelle et al. (2010a, b) used a nonlinear model to investigate the influence of inner sandbar on the outer one and vice-versa. The work is of particular relevance because it proposes a novel

framework to analyse the coupling that moves beyond the traditional 'template' versus 'self-organization' debate (Coco and Murray, 2007). The computations of Castelle et al. (2010a, b), and also others like Price et al. (2013), start from an initial bathymetry characterized by a double bar systems with a crescentic bar superimposed to the outer bar. The use of this type of initial configuration favours the growth of crescentic shapes in the inner bar, with the same wavelength as the one in the outer bar, which originally might have developed through self-organization. This authors named this phenomena 'morphological

coupling'. In terms of physical processes, the contributions by Castelle et al. (2010a, b) address the role of breaking-induced (dominant for large spacing of the crescents or strong breaking conditions) versus friction-induced circulation (dominant for





small spacing of the crescents). This balance induces the emergence of patterns that in broad terms are 'in-phase' when wave focusing by refraction is dominant and 'out-of-phase' when breaking-induced circulation is the primary flow driver.

Overall, it appears that numerical studies have extensively explored the sandbar response to offshore wave characteristics but, aside from the initial study by Brivois et al. (2012) that analyzed the stability of two different beach profiles at Truc Vert beach (France), have not attempted to systematically study the role of initial bathymetry on the evolution of the double sandbar systems. Here, we use a numerical model based on linear stability analysis, MORFO62 (Ribas et al., 2012), to study the combined role of hydrodynamic conditions and initial cross-shore sandbar profile on the evolution of double sandbar systems. The different emerging patterns are then characterized. Special attention is devoted to distinguishing when the emerging patterns evolve autonomously (an individual sandbar) or when they are truly the result of morphological coupling (both sandbars interacting with each other).

## 2 Numerical model

The numerical model describing the surf zone hydrodynamics is based on the depth- and time-averaged momentum and continuity equations coupled to the wave-energy and phase equations. The momentum balance and water mass conservation equations read

$$\frac{\partial v_i}{\partial t} + v_j \frac{\partial v_i}{\partial x_j} = -g \frac{\partial z_s}{\partial x_i} - \frac{1}{\rho D} \frac{\partial}{\partial x_j} \big( S_{ij}^W + S_{ij}^R - S_{ij}^t \big) - \frac{\tau_{bi}}{\rho D}$$

$$\frac{\partial D}{\partial t} + \frac{\partial}{\partial x_j} \big( D v_j \big) = 0 \quad , \; i,j = 1,2 \tag{1}$$

In this notation, the Einstein convention is adopted, i.e. if an index appears twice in a term we assume a summation over that index. Here, the vector $\boldsymbol{v}(x_1, x_2, t)$ is the wave- and depth-averaged mass flux current ($\boldsymbol{v} = (v_1, v_2)$), $t$ is time, $x_i$ indicates the horizontal spatial coordinates ($x_1$ and $x_2$ are the cross-shore and alongshore directions), $g$ is gravity, $z_s$ represents the mean sea level, $\rho$ is the water density, $D$ is the total mean depth, $S_{ij}^W$ and $S_{ij}^R$ are the radiation stresses due to waves and rollers, while $S_{ij}^t$ represents the turbulent Reynolds stresses. Finally, $\tau_{bi}$ indicates the bed shear stress.

The wave energy balance equation reads

$$\frac{\partial E}{\partial t} + \frac{\partial}{\partial x_j} \big( (v_j + c_{gj}) E \big) + S_{jk}^W \frac{\partial v_k}{\partial x_j} = -\mathcal{D}^W \quad , \; j,k = 1,2 \tag{2}$$

where $E = \frac{1}{8} \rho g H_{rms}^2$ is the wave energy density, with $H_{rms}$ being the root mean squared wave height, $c_{gj}$ are the components of the group velocity and $\mathcal{D}^W$ represents the wave energy dissipation rate due to wave breaking. The roller energy balance equation reads

$$\frac{\partial}{\partial x_j} \big( 2(v_j + c_j) R \big) + \mathcal{S}_{jk}^R \frac{\partial v_k}{\partial x_j} = -\mathcal{D}^R + \mathcal{D}^W \quad , \; j,k = 1,2 \,. \tag{3}$$





$R$ is the energy density of the rollers, $c_j$ are the components of the phase velocity and $\mathcal{D}^R$ represents the wave energy dissipation rate due to the rollers. The wavenumber $\boldsymbol{K}(x_1, x_2, t)$ ($\boldsymbol{K} = (K_1, K_2)$) of the waves obeys the equation

$$\sigma + v_j K_j = \omega \qquad \sigma^2 = g|\boldsymbol{K}|\tanh(|\boldsymbol{K}|D) \tag{4}$$

where $\sigma$ and $\omega$ are the intrinsic and the absolute wave frequencies, respectively. The wave energy dissipation rate is parameterized using the formulation by Church and Thornton (1993):

$$\mathcal{D}^W = \frac{3B^3 \rho g \sigma H_{rms}^3}{32\sqrt{\pi}D} \left(1 - \left(1 + \left(\frac{H_{rms}}{\gamma_b D}\right)^2\right)^{-2.5}\right) \left(1 + \tanh\left(8\left(\frac{H_{rms}}{\gamma_b D} - 1\right)\right)\right) \tag{5}$$

in which $B$ ($B^3 = 2.2$) is a parameter describing the type of breaking, $\gamma_b$ ($= 0.42$) is the expected saturation value of $H_{rms}/D$. The roller energy dissipation rate is modeled following Ruessink et al. (2001):

$$\mathcal{D}^R = \frac{2gR\sin(\beta_{\text{rol}})}{c}, \tag{6}$$

where $\beta_{\text{rol}}$ ($\leq 0.1$) is the angle of the wave/roller interface. Wave radiation stresses, stresses due to roller propagation and

turbulent Reynolds stresses (Svendsen, 2006) are expressed as

$$\mathcal{S}_{ij}^W = E\left(\frac{c_g}{c}\frac{K_i K_j}{K^2} + \left(\frac{c_g}{c} - \frac{1}{2}\right)\delta_{ij}\right)$$

$$\mathcal{S}_{ij}^R = 2R\frac{K_i K_j}{K^2}$$

$$S_{ij}^t = \rho \nu_t D\left(\frac{\partial v_i}{\partial x_j} + \frac{\partial v_j}{\partial x_i}\right) \quad , i,j = 1,2 \tag{7}$$

where $\delta_{ij}$ is the Kronecker delta symbol and the turbulent kinetic diffusivity is

$$\nu_t = M\left(\frac{\mathcal{D}^W}{\rho}\right)^{\frac{1}{3}} H_{rms} \tag{8}$$

with $M$ a parameter of $O(1)$ that characterizes the turbulence. With respect to shear stresses, we use a linear friction law $\tau_{bi} = \rho\mu v_i$ ($i = 1,2$) with $\mu = \left(\frac{2}{\pi}\right) c_D u_{rms}$. We model the drag coefficient as

$$c_D = \left(\frac{0.40}{\ln(D/z_0) - 1}\right)^2 \tag{9}$$

where $z_0$ is the bed roughness and $u_{rms}$ is the root mean square wave orbital velocity at the edge of the wave-induced boundary layer:

$$u_{rms} = \frac{H_{rms}}{2}\frac{g}{c}\frac{\cosh(|K|z_0)}{\cosh(|K|D)}. \tag{10}$$



To simulate morphological evolution the hydrodynamic field must be coupled to a sediment transport formulation and to the conservation of sediment mass. Bed evolution is described as

$$\frac{\partial z_b}{\partial t} + \frac{1}{1-p}\frac{\partial q_j}{\partial x_j} = 0 \quad , j = 1, 2 \tag{11}$$

where $z_b$ represents the mean sea level, $p\,(=0.4)$ is the porosity of the seabed and $q_j$ are the components of the volumetric sediment transport whose parameterization is given by the Soulsby–van Rijn formula (see Soulsby, 1997), expressed in the form

$$q_i = A_s(u_{\text{stir}})^{2.4}\left(v_i - \gamma\, u_{\text{stir}}\frac{\partial h}{\partial x_i}\right) \quad , i = 1, 2 \tag{12}$$

where $A_s$ depends on the sediment properties and $\gamma$ is a bedslope coefficient. The term $A_s(u_{\text{stir}})^{2.4}$ is the depth-integrated sediment concentration ($C_{di}$) Following Ribas et al. (2012), $u_{\text{stir}}$ is a stirring velocity that takes into account the depth-averaged currents, the wave orbital velocity and roller-induced turbulence velocities.

The system of equations, when alongshore uniformity is assumed, allows for a state of morphodynamic equilibrium (steady state) for the hydrodynamic forcing conditions. The solution of the system of equations is perturbed and equations linearized as in any standard linear stability analysis (Dodd et al., 2003; Calvete et al., 2005). For a given set of forcing conditions (wave height and period; normal incidence is assumed throughout this study) and cross-shore profile, outputs of the analysis are the characteristics of the fastest-growing instability of the system: the growth time (the e-folding time) and the alongshore pattern periodicity (herein indicated using the alongshore wavenumber $k$). Boundary conditions and more details about the numerical model can be found in Ribas et al. (2012) and in Calvete et al. (2005) .

## 3 Results

We initially present an example of the model analysis for a specific bathymetry (alongshore uniformity of the initial cross-shore profile is considered) and offshore wave conditions. For this case, we use a significant wave height $H_{rms} = 1.5$ m and a wave period $T = 10$ s with normally incident waves. The first step of linear stability analysis is evaluating the equilibrium state, which represents the morphodynamic equilibrium previously discussed, of the equations presented in the previous section considering a fixed seabed. We assume that bathymetry of equilibrium state is characterized by an evolution that occurs over a long temporal scale compared with the growth of the emerging morphological pattern.

Figure 2 shows the bottom cross-shore profile which is characterized by the presence of two sandbars with crests at about 200 and 480 m in the cross-shore direction, with the distance between the sandbar crests $\Delta x = 280$ m, and a difference of about $\Delta D = 2.5$ m between the water depths. The other panels show other characteristics of the hydrodynamic and sediment transport (for example, notice the effect of the sandbar on wave transformation). The basic state, different for different cross-shore beach profiles, is then perturbed and possible emerging modes are analysed in terms of their growth rate. Figure 3 shows the growth rates for the example being analysed. Three different modes are present with the fastest growing one, mode 1, characterized by an alongshore spacing close to 420 m (the wavenumber is about 0.015 m$^{-1}$). The second and third modes are characterized by slower growth rates and an alongshore spacing close to 170 and 500 m, respectively.





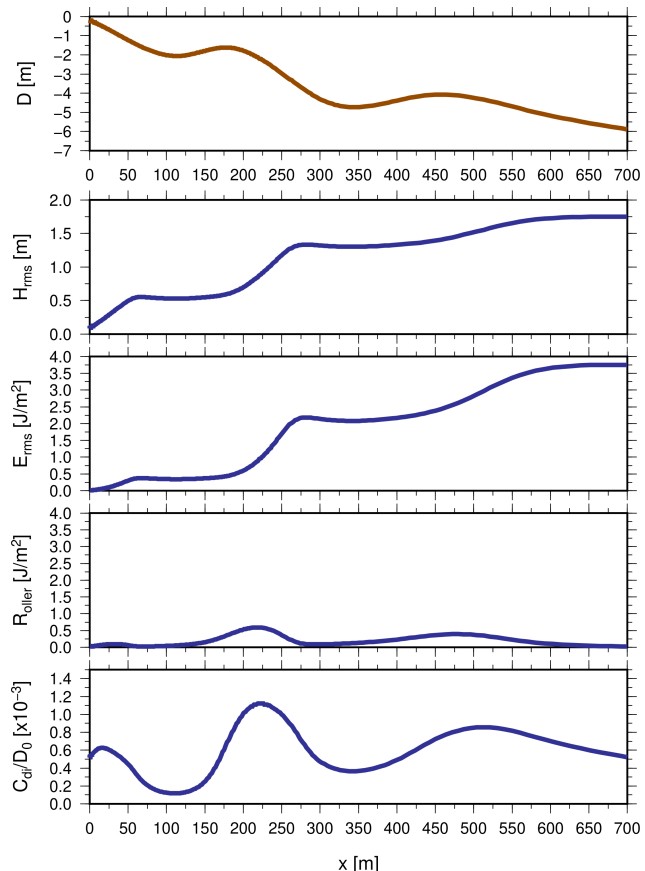

**Figure 2.** Basic state variables for a typical cross-shore beach profile (shoreline is at $x = 0$). From top to bottom: water depth, root mean square height, wave energy dissipation, roller energy and depth-averaged sediment concentration.

The water depth and circulation pattern associated to the fastest growing value of each of the three modes are shown in Figure 4. The patterns display some evident differences with respect to which of the two sandbars is unstable. The mode 1 represents the classic crescentic sandbar instability and only the inner sandbar is unstable. Circulation over the inner sandbar consists of onshore flow over the shoals and offshore flow in the lower/channel areas consistent with the traditional mechanism of crescentic sandbar or rip channel formation (Falqués et al., 2000; Calvete et al., 2005). The mode 2 displays instead an instability that comprises the zone between the inner sandbar and the shoreline. Circulation and morphology develop also close to the shoreline in the form of transverse bars aligned to the lower/channel areas of the inner sandbar crescents. Finally, the mode 3 shows an instability of the outer sandbar with small in-phase signatures on the inner sandbar. The growth rate of the different modes can be understood following Ribas et al. (2015). The pattern related to the fastest growing mode, mode 1, arise in the areas of more intense dissipation of wave energy (both in wave and roller energy, Figure 2) and where the gradients in depth-averaged sediment concentration are larger (Figure 2). Similarly, mode 2 is associated to an instability extending close



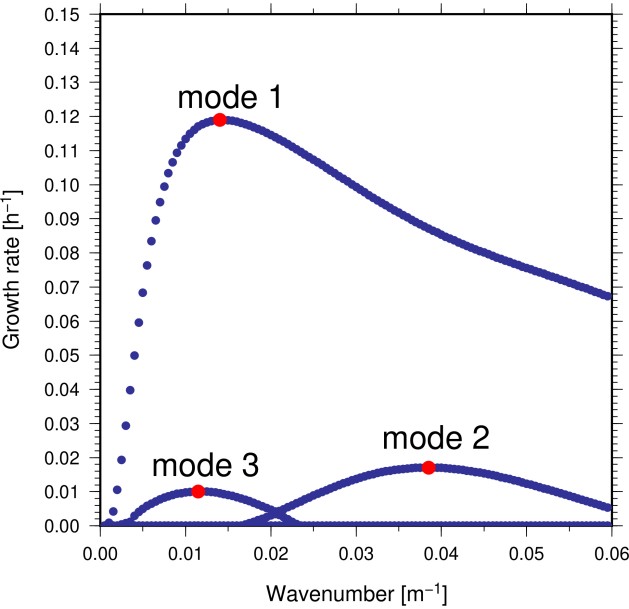

**Figure 3.** Growth rates as a function of the wavenumber for $H_{rms} = 1.5$ m and $T = 10$ s. The bathymetry considered in this case is the same as presented in Figure 2.

to the shoreline where the gradient in depth-averaged concentration leads to the development of transverse sandbars associated to an offshore flow (Ribas et al., 2015). Mode 3 is characterized by less intense circulation and depth-averaged concentration which extends to the inner sandbar.

This same approach has been applied to a series of different bathymetries to study the effect of the distance, $\Delta x$, and difference in water depth, $\Delta D$, between the two sandbars. Figure 5 shows the series of cross-shore profiles that will be considered in this study. We have tried to isolate individual effects and for example, profiles in red will specifically address the sensitivity to the difference in water depth between the two sandbars. Similarly, profiles in blue will directly assess the role of the distance between sandbar crests.

Applying linear stability analysis to the beach profiles shown in Figure 5 results in a variety of beach responses, each identified by a specific mode. The patterns that are predicted to emerge vary largely and we have attempted to group them according to their characteristics. In Figure 6 we show the different patterns obtained and group them in terms of which sandbar is unstable and the type of coupling occurring between sandbars. We use the letters I and O to indicate patterns that are associated only to the inner or offshore sandbar, respectively. The symbols + and - are used to indicate possible 'in-phase' or 'out-of-phase' coupling so that overall, a pattern indicated with the symbols O+ refers to a configuration where the dominant

effect of the instability is over the offshore sandbar (letter O) while the inner sandbar shows some limited 'in-phase' coupling (symbol +). When the coupling between sandbars is obvious, we denote the patterns with the letters IO adding the symbol + or - depending on whether the sandbars show 'in-phase' or 'out-of-phase' coupling. Finally, just as shown in Figure 4 for mode 3,





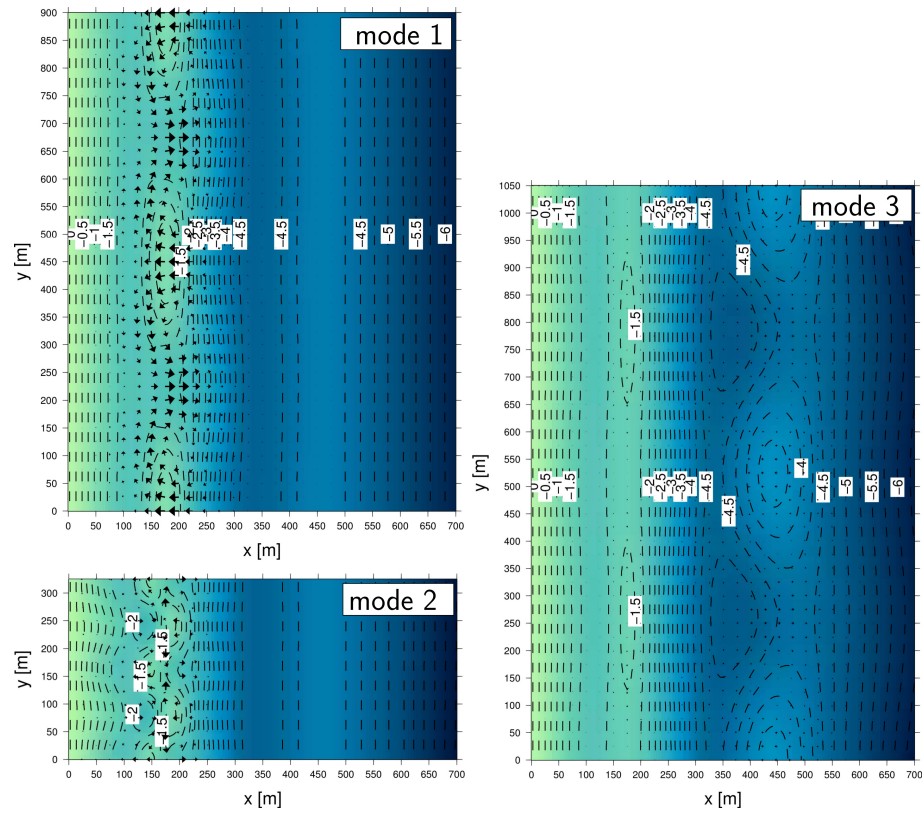

**Figure 4.** Fastest growing modes for the peaks in the growth rates shown in Figure 3. Shoreline is on the left of each plot and labels indicate water depth.

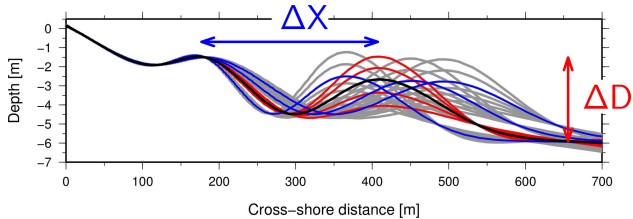

**Figure 5.** Geometry of cross-shore beach profiles used in this study. Different colours are used to highlight that some of the profiles were specifically designed to analyse the effect of variations in the distance or in the relative depth between sandbar crests.

several unstable configurations that also involve changes close to the shoreline. In the remaining of the manuscript, the possible
effect on shoreline/inner surfzone morphology has been indicated using the subscript s.

Earth **Surface**
**Dynamics**
Discussions

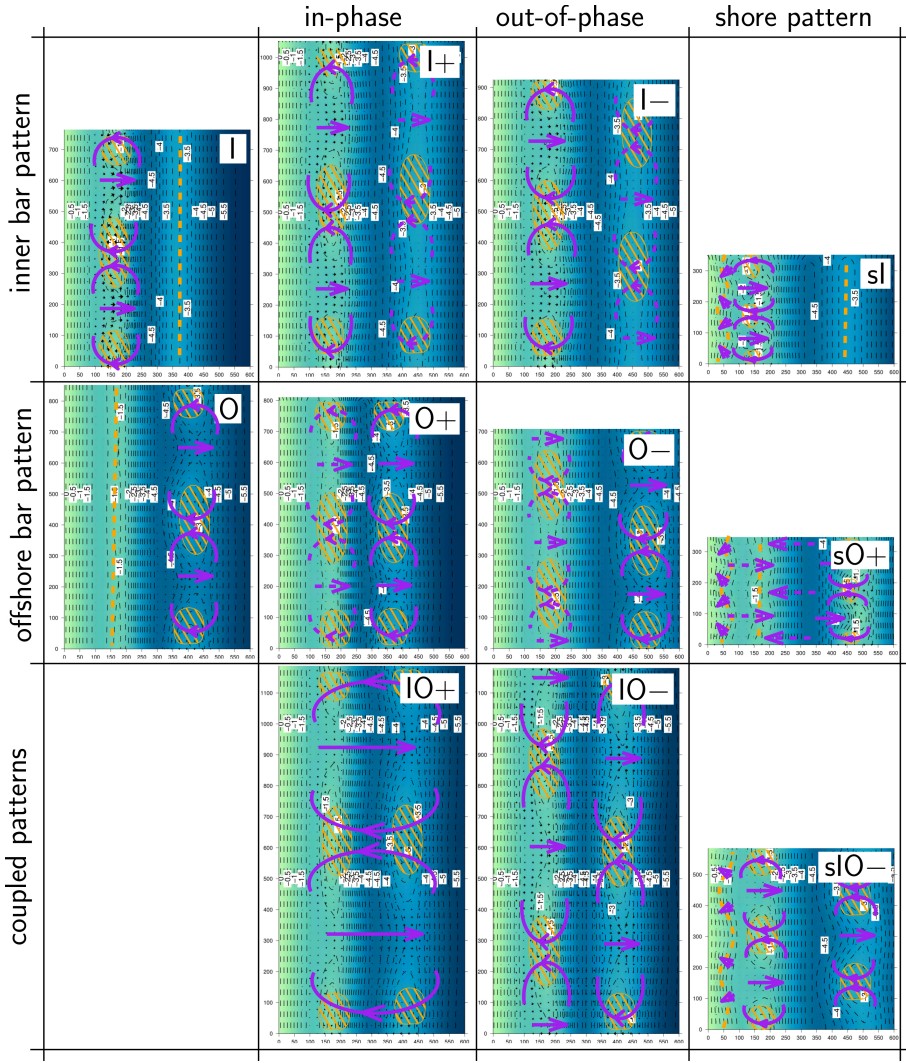

**Figure 6.** Unstable morphological patterns obtained in this study. The letters I and O indicate a dominance of the inner and outer sandbar, respectively. Patterns indicated with the code IO respresents modes where both sandbars are simultaneously unstable. The symbols + and - are used to indicate possible 'in-phase' or 'out-of-phase' coupling. Modes affecting shoreline/inner surfzone morphology have been indicated using the subscript s.

We have run simulations over the bathymetries presented in Figure 5 using three different values of wave height (equal to 1.0, 1.5 and 2.0 m) and keeping the wave period fixed (equal to 10 s). Results are presented in Figure 7 and 8. Figure 7 shows the emerging modes as a function of wave height and sandbar distance; Figure 8 shows the corresponding growth rates and spacing. Three unstable modes are usually present but when wave height is smallest ($H_{rms} = 1.0$ m), only 2 modes are

unstable. The first mode, the fastest growing, displays a similar pattern for the three values of the wave height considered.

Earth **Surface**
**Dynamics**
Discussions

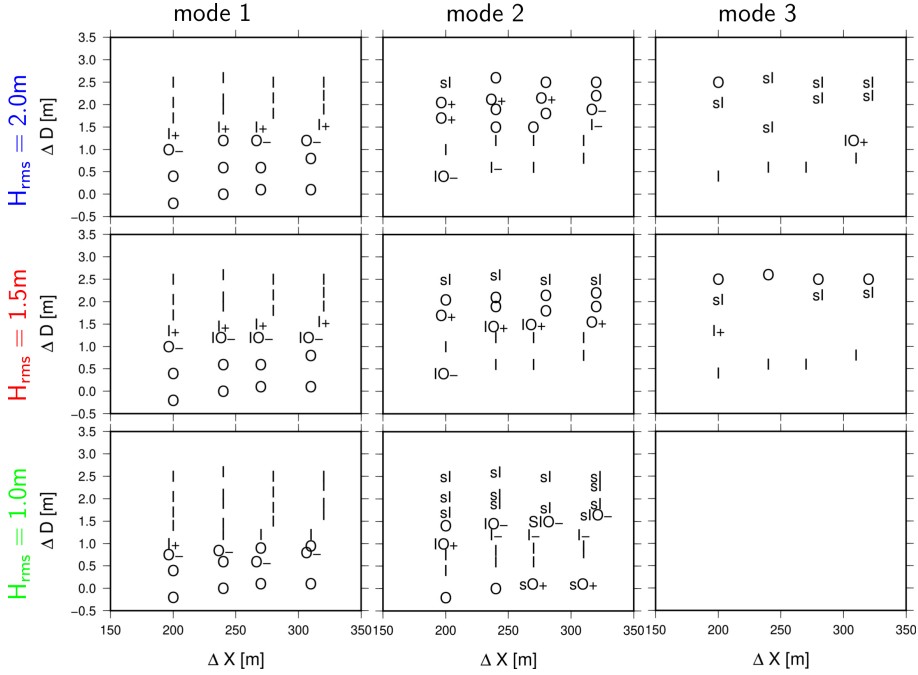

**Figure 7.** Unstable morphological patterns as a function of $\Delta D$ and $\Delta x$. Each code represents a different pattern (detailed in Figure 6). The top, center and bottom panels represent results obtained for wave height equal to 1.0, 1.5 and 2.0 m.

When the difference between the sandbar crests, $\Delta D$ is large, the fastest growing mode is type I which implies that the inner sandbar is unstable and develops into a crescentic shape. Because of the large difference in water depth between sandbar crests, the offshore sandbar is essentially inactive while when $\Delta D$ is small most of the wave breaking is concentrated on the offshore sandbar which is likely to go unstable and develop crescents (type O). For intermediate differences in the water depth between the sandbar crests, a transition from type I to O can be observed. In most cases the transition occurs through the development of an I+ pattern (the instability is stronger at inner sandbar and the outer sandbar reflects limited 'in-phase' coupling). As $\Delta D$ decreases an instability of type O- is also likely to develop (the instability is stronger at outer sandbar and the inner sandbar reflects limited 'out-of-phase' coupling). For $H_{rms} = 1.5$ m the transition also results in the development of fully coupled patterns, type IO. While the patterns show an evident dependence on $\Delta D$, the role of $\Delta x$ on the emergent unstable patterns is extremely limited (Figure 7). The second mode, characterized by lower growth rates, is often specular to mode 1 (i.e. if for a particular combination of $\Delta D$ and $\Delta x$ the mode 1 instability is of type I, for the mode 2 the instability is type O). It is also worth pointing out that no mode 1 configuration affects shoreline morphology while modes associated to changes at the shoreline appear more frequently as mode 2 and 3 especially if $\Delta D$ is large.

In order to understand the underlying differences between the IO modes and the modes I or O, additional experiments have been carried out. For example, simulations for which modes I+ or I- are found, were repeated but without sediment transport in

Earth **Surface**
**Dynamics**
Discussions

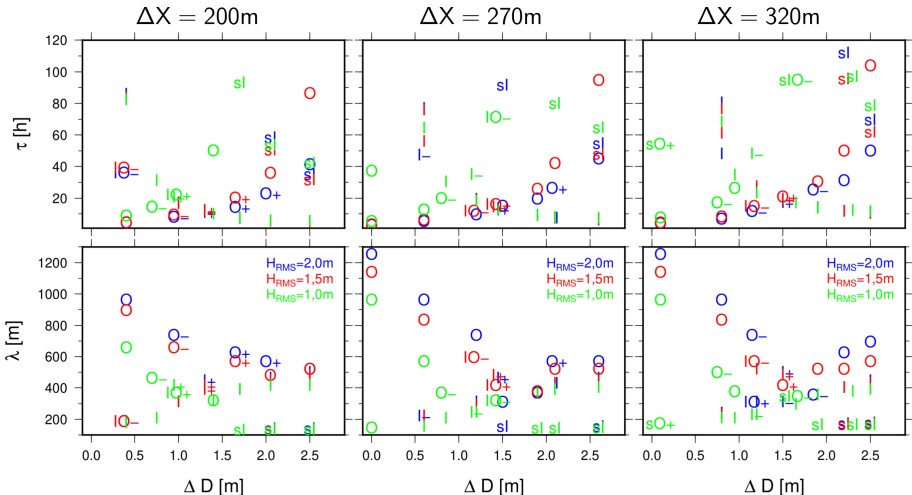

**Figure 8.** Growth time (top) and rip channel spacing (bottom) as a function of $\Delta D$. The left, center and right panels represent results obtained for $\Delta x$= 200, 270 and 320 m, respectively. Blue, red and green symbols refer to wave height equal to 1.0, 1.5 and 2.0 m.

the outer bar. As a result, modes with similar growth rates and spacing were found but with no extension on the outer sandbar. Same results were found in the equivalent experiments for O+ and O-. For conditions leading to modes IO, cancelling the sediment transport in any of the two sandbars produced modes limited to a single bar with significant differences in the growth rate and spacing. Modes IO should be then considered as a mode that develops affecting the two sandbars simultaneously.

Given that the sensitivity to $\Delta x$ is limited, we fixed its value (equal to 200, 270 and 320 m) and specifically looked at the growth rates and spacing (Figure 8). Results can be interpreted by looking at the trends of the individual type of patterns. For example, independently of the value of wave height, patterns of type I consistently show a marked decrease in the growth time with increasing $\Delta D$ (see top panels in Figure 8). The decrease in growth time is accompanied by a moderate increase in the spacing of the rip channels (the spacing of mode I instabilities never exceeds 400 m). The same behaviour is observed for all

distances between the sandbar crests considered in this study. Mode O shows an almost opposite behaviour: the growth time increases with $\Delta D$ while the spacing of the rip channels diminishes (for $\Delta x = 320$ m a slight increase in spacing is observed for very large $\Delta D$). The largest rip channel spacing observed for mode O is in excess of 1,200 m which is about twice the largest spacing observed for mode I. Finally, the unstable modes that have a shoreline signature are all characterised by large values of $\Delta D$, large growth times and short spacing (about 200 m).

**4   Discussion**

We focused on the morphodynamics of double sandbar systems and tried to investigate under which conditions the system is unstable to perturbations ultimately resulting in the development of surfzone patterns like rip channels/crescentic sandbars. We





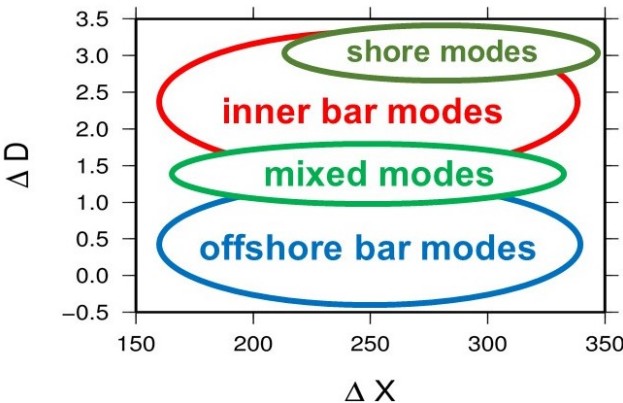

**Figure 9.** Sketch of the most likely fastest growing modes as a function of the geometry of the cross-shore profile. $\Delta D$ represents the difference between the water depth over the two sandbar crests while $\Delta x$ is the distance between the two sandbar crests.

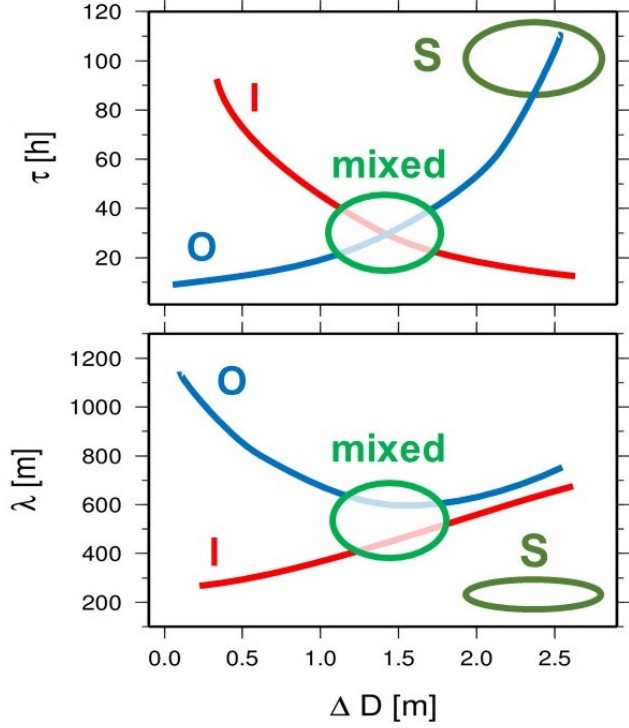

**Figure 10.** Sketch summarizing findings in terms of (top panel) growth time and (bottom panel) crescentic sandbar spacing.





use linear stability analysis to discover the morphological configurations that can arise as a result of the feedbacks between hydrodynamics, sediment transport and morphological change. We primarily focused on the sensitivity to the initial seabed

cross-shore profile, varying the distance between sandbar crests or varying the difference between the water depth over the two sandbar crests. We generally observe large variability in the response of the system to changes in bathymetric details. This is not entirely unexpected since the amount of wave breaking induced by the geometry of the outer sandbar is critical to determine if the two sandbars are coupled or if only one of the two sandbars can be unstable. This is in agreement with the findings by Castelle et al. (2010a) reporting that the type of horizontal flow circulation over the outer sandbar (driven either by

refraction or by wave breaking) is ultimately responsible for the possible coupling between sandbars. For this reason, while the distance between the sandbar crests is unimportant in determining patterns and trends (Figure 7), the difference in water depth is a critical parameter to determine the shape and characteristics of the fastest growing model. In our model, a large difference in the water depths over the two sandbar crests (e.g. $\Delta D = 2.5$ m) implies that limited wave set-up and breaking occur over the outer sandbar which is essentially inactive. In this case the fastest growing mode is always related to the inner sandbar which

is likely to behave as a single sandbar system and become unstable (Figure 9) following the physical mechanisms described by Calvete et al. (2005). When the difference between the sandbar water depths is small (e.g. $\Delta D = 0.5$ m) strong wave breaking occurs over the outer sandbar and the fastest growing instability is related to the outer sandbar (Figure 9). Coupling between the two sandbars occurs for intermediate differences in the water depth of the sandbar crests while the presence of unstable configurations that involve the shoreline, only occurs for the largest water depth difference (Figure 9). This behaviour is also

evident when looking in detail at the spacing of the emerging rip channel pattern and at the growth time of the unstable mode (Figure 10). As synthesised in Figures 9 and 10, results bear little dependency on $\Delta x$ and the overall behaviour of the system is governed by $\Delta D$ and $H_{rms}$. When $\Delta D$ is small, the presence and characteristics of an unstable mode depend on the value of $H_{rms}$. As shown in Figure 10, for small $\Delta D$ the outer sandbar spacing depends on $H_{rms}$ but tends to be large, while for large $\Delta D$ the dependency of the spacing to $H_{rms}$ is smaller. Inner sandbar modes dominate instead for large $\Delta D$ and small

$H_{rms}$.

Regarding the morphological coupling discussed by other authors (Castelle et al., 2010a; Price et al., 2014), our results derived from linear stability analysis can distinguish between modes that develop in one of the bars and that force an instability over the other sandbar. At the same time, we obtain modes that develop simultaneously over the two sandbars. In the first case, we interpret that there is a primary mode affecting one of the sandbars with the other sandbar evolution being passively

slaved to its morphodynamics. In the second case, the instability developing over the two bars is related to the same mode and, therefore, the emerging pattern shows full sandbar coupling. This full morphodynamic coupling occurs for intermediates differences of sandbar depth. For small differences of depth, instabilities on the outer bar dominate, whilst for larger differences of bars depths the main instability is located at the inner bar (although the wavelength of the crescentic bars on the inner and on the outer bar appear to be very similar). The transition from forced to fully coupled occurs smoothly for the parameter space

examined. Since the present model is a linear model, the concept of coupling is limited to the initial morphological formation and not to the subsequent nonlinear interaction which might lead to coupling over longer time scales (days to weeks). For





cross-shore profiles that allow for large wave energy to reach the shoreline, the model predicts the formation and coupling of shoreline patterns even though the model does not include swash dynamics and we considered a fixed shoreline.

Despite our attempts to provide a detailed description of hydro- and morphodynamics, the numerical model remains sim-

plified and does not include a number of physical processes that in the context of surfzone morphodynamics can be relevant. As for the case of many surfzone morphodynamic studies, hydrodynamic forcing is simplified and the effect of directional and frequency spread in the wave field as well as tidal variations are all neglected. One could expect that the primary effect related to these processes was a decrease in the growth time of the features without necessarily affecting the type of morphodynamic patterns predicted to grow. We also neglected the role of wave angle (we only considered normally approaching waves) which

has been shown to be relevant for the coupling of sandbar systems (Price and Ruessink, 2011; Price et al., 2013). On the other hand, we include a detailed modelling of the effect of wave-induced rollers that has been shown to be important for the development of surfzone features (Ribas et al., 2011; Calvete et al., 2012) but whose effect on double sandbar systems had not been considered before. Finally, the study does not address some of the possible effects on sediment transport associated to undertow and wave asymmetry which is nonlinear and, particularly for varying cross-shore beach profiles, could quantitatively affect the

results. Despite these shortcomings, the model reproduces morphodynamic patterns which are consistent with the presence of coupled sandbar patterns. Although the objective of this contribution is limited to a numerical analysis of the possible unstable patterns arising in double sandbar configurations, model predictions are in qualitative agreement with observations of the Truc Vert (France) double sand bar system (Brivois et al., 2012). Lack of detailed and systematic measurements of bathymetric evolution of coupled sandbar systems remains the biggest obstacle to model testing in this area of research.

Our findings have clear implications for the understanding of observed coupled sandbar patterns. Coupled sandbar systems are usually considered as the result of one sandbar affecting another. Our results indicate that coupling can also emerge as a result of single unstable mode. The apparent differential growth of each sandbar might lead to think one sandbar is forcing the coupling over the other sandbar. Our results indicate that a coupled pattern, with perturbations over each sandbar of different amplitude, can also arise without invoking one sandbar as a forcing mechanism. In addition, our results indicate that a variety of

modes can grow for similar conditions. Although we do not deal with the nonlinear behaviour of the patterns, one can envisage that growth and interaction between multiple modes can become a source of spatial variability in the observed pattern.

## 5  Summary

In order to gain insight on the physical processes that govern the development of coupled sandbar patterns we have analysed the linear stability of a system of equations describing the morphodynamics of a double sandbar system. Our results indicate

the development a variety of morphological configurations where inner and outer sandbar show 'in-phase' and 'out-of-phase' coupling, or no coupling. Our study points at the combined influence of offshore wave characteristics and initial cross-shore bed profile in determining the alongshore wavelength and growth rate of the fastest growing mode/pattern. Overall, inner bar-modes are dominant for large differences between the two water depth of the sandbars and small offshore wave heights while patterns related to the outer sandbar dominate for small values of the difference in sandbar depths. For intermediate differences



between the two sandbars depths, patterns on both longshore bars appear to be fully coupled. Relatively larger waves and large depth over the outer sandbar can induce patterns close to the shoreline/inner surfzone. Although initial comparisons appear to support our modelling, continued model development, particularly trying to address the effects of cross-shore wave-induced sediment transport, remains critical to improve understanding and predictability of these natural systems.

*Author contributions.* Coco and Calvete designed the study from inception to dealing with the execution and the analysis of the numerical experiments, and the writing of the first draft of the manuscript. Ribas, de Swart and Falques contributed to subsequent drafts.

*Competing interests.* None

*Acknowledgements.* This research was funded by the Spanish Government (MINECO/FEDER) grant number CTM2015-66225-C2-1-P. GC funded by a NZ Hazard-Platform grant (GNS-MBIE C05X0907).



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
