# Peer review of "Emerging crescentic patterns in modeled double sandbar systems under normally-incident waves"

_Earth Surface Dynamics, 2019_

## Referee Comment (RC1) · Anonymous Referee #1 · 2 Jan 2020

**General comments:**

The manuscript "Emerging crescentic patterns in modeled double sandbar systems" by Coco et al. is based on a systematic linear stability analysis of the development of crescentic patterns at double-barred beaches. The authors show that the cross-shore distance and particularly the crest depth difference between the inner and outer bars is critical to occurrence and type (out of phase vs in phase) of coupling and preferred spacing of the morphodynamic instabilities. I really enjoyed reading this manuscript and I thank the Editor for inviting me to review this one. Although there has been quite a bit of work published on the modelling and observation of coupling patterns of double-barred systems, including mine, this work provides a wealth of interesting material and new insight into the influence of both wave conditions and mean profile

characteristics on emerging coupling patterns. I therefore recommend publication after some modifications have been made. My comments are reasonably minor to moderate and should be straightforward to address, see below.

**Specific comments:**

Abstract: The sentence "For intermediate differences between the two sandbars depths, patterns on both longshore bars appear to be fully coupled" was quite unclear to me before going through the manuscript, and it should therefore be slightly rephrased as by "fully coupled" the author mean something more like the 2 bar patterns grow at a similar rate.

Introduction: This is a very nice section providing background on coupling patterns. I think, however, that the authors should make clearer that nothing has been done on the influence of the distance between the bars, and crest depth difference (although for the latter it is tentatively said), in other words, the authors may put more emphasis onto what is new in their contribution. I was not comfortable with the use of the word 'geometry' in this section. To me the sandbar geometry refers to the barline (2D horizontal) and does not include depth, I would rather talk about 3D morphology, but I am not English native so my comment may not be relevant. The authors should also refer to the work of Garnier et al (2013, GRL) when dealing with bar straightening under obliquely incident waves. I also recommend to indicate that the authors will stick to shore-normally incident waves in the last paragraph of the introduction section.

Numerical Model: Please double check all notations: for instance L142 zb is not the mean sea level (zs in eq (1)) but the mean seabed elevation, h in eq(12) is not defined it should be the bathymetric perturbation (deviation from the basic state). It would be nice to add a short paragraph with the equation of the perturbation and indication of how \tau is computed (why referred to as growth time rather than e-folding time in most papers?). I understand that this is given in earlier papers, but that would help the reader to have a standalone paper.

Results: * I encourage to modify Fig. 4 to improve readability of arrows and perturbation with for each mode the left-hand panels water depth with iso-contours and right-hand panels perturbation h (not contoured) with currents. Please also indicate the time at which the different bathymetries have been plotted. * L198: "When the coupling between sandbars is obvious" I guess that the authors used some kind of more or less objective threshold in terms of ratio of perturbation amplitude at the inner and outer bar to discriminate between "obvious" and more "subtle" couplings, please clarify. * L200: the same applies here, did the authors use a some kind of threshold in term of perturbation near the shoreline, e.g. at a given basic state iso-contour ?

Discussion: * Dealing with the limitations of the study is half of the discussion, I advise slightly to shorten the limitations, which should start as a new paragraph L270, and/or extend the first part of the discussion * In the limitations part, the authors may add a couple of sentences on the fact that coupling at half of the outer bar wavelength (Castelle et al., 2010b) cannot be reproduced here. * My own 'empirical' knowledge of double barred beaches I've been to along different coasts is that out-of-phase coupling is much more common than in-phase coupling, this also applies to shoreline-sandbar coupling along single barred beaches. The numerical results here indicate the couplings are about equally distributed between in-phase and out-of-phase. I do not necessarily ask the authors to discuss this, because my qualitative observations may be biased and they may not think the same, but I am curious to know what the authors think about the coupling type predominance and potential mismatch with model outputs.

**Technical corrections:**

The authors may consider adding 'linear stability analysis' and/or 'under shore-normally incident waves' in their title.

The paper is written in very good English, however there are a few typos here and there (L116 missing bracket, remove comma at the end of equation (6), add '.' before

'Following' L147, uncapitalize 'X' of the 3 top \Delta X in Fig. 8, idem in Fig. 7, ...) and I recommend a very last proofread.

Remove or increase label size in Fig. 6 (cross-shore/longshore distance, iso-contours)
* * *

---

## Referee Comment (RC2) · Anonymous Referee #2 · 22 Feb 2020

Coco et al 2020 investigate the stability of double-barred beaches using a numerical model based on linear stability analysis. They focus on the development of morphological patterns in the two nearshore bars, expressed through the development of alternating shallower and deeper areas along the initial position of the bar crests. Building upon various findings from the literature, they choose to vary specific initial morphological and hydrodynamic boundary conditions to further investigate the variability in morphological coupling between both bars. The authors find that the depth difference between the bars determines the resulting coupling modes, with variations in wave heights playing a secondary role.

GENERAL COMMENTS

The paper provides novel and interesting insights into the variable development of mor-

phological coupling in nearshore zones. The role of the bar depth on this difference in coupling development is particularly interesting, providing an interesting analysis pathway for future modelling and field measurements of morphological coupling. However, I do feel that the authors could provide the reader with some further guidance regarding the motivation for the aim of this study (in the introduction) and the framing of their results within existing body of literature on the topic (in the discussion section). As such, before publishing, I recommend minor-moderate revisions. I elaborate on my reasoning and provide further suggestions below.

Besides elaborating on the motivation for the study and embedding of the results, it would be insightful for the reader if these authors in particular (given their shared experience with this type of model and other modelling approaches) could provide a (brief) reflection as to why LSA is particularly suitable for tackling this study. Would the use of, for example, a nonlinear model lead to similar conclusions regarding the emergence of the patterns? Why (not)? The discussion section includes a reflection on the use of LSA herein (L274-289), but please mention in the methodology section what makes LSA suitable for answering the research question.

The paper reads very well and is equally well-structured. Some minor textual corrections are provided.

OTHER COMMENTS

Abstract: Please include the aim, or knowledge gap, in the abstract. The abstract now starts with a description of the approach, followed by the description of the results in Line 5.

L9 it is unclear what is meant by "inner bar-modes are dominant" -> Please describe what "inner-bar modes" are, and also what other modes there are.

L92-96 Somewhere here, when introducing the use of the model, elaborate on the reason for opting for a model based on LSA.

L177 Transverse bars are mentioned here for the first time -> Please mention these in the introduction section as features that may appear coupled to the sandbar pattern (Ribas et al 2014, Ocean Dynamics).

L185-189 This paragraph belongs in the methods section, including figure 5. The choice for focusing on delta(D) and delta(x) should be elaborated upon, probably in the introduction ( somewhere in L78-92). Why not, for example, investigate the effect of changing the cross-shore slope (which, admittedly, inherently includes changes in cross-shore distance and bar depth, but also bar volume)? For sake of clarity, it is also worth noting that delta(D) here means changing the depth of the outer bar, while keeping the inner bar depth the same.

L287-288 Here it is mentioned that "model predictions are in qualitative agreement with observations of the Truc Vert double sandbar system". How do they agree? Please explain or show by means of a comparitive figure.

L288-289 Here is mentioned that bathymetries of coupled sandbars are scarce, obstructing the comparison of the model with field observations. Could a general comparison of measured delta(D) give some insight into the validity or probability of the model results, or do you expect delta(D) to differ when bars couple? How do the findings of this study translate to future field studies? Please reflect on this.

L271-273 (and elsewhere) The model shows that large waves lead to a shoreline that couples to the outer bar. Does this correspond to the observation of coupling between shoreline embayments and the outer bar shape during a severe storm, by Castelle et al (2015)?

L293-294 "Our results indicate .. single unstable mode." -> This is indeed a key point of this study. This statement would be even stronger if it was posed as the problem or hypothesis you wish to tackle with this study (also see my comment above).

TECHNICAL CORRECTIONS

L9 two sandbars crests -> two sandbar crest

L84 THESE authors named this PHENOMENON

L93 hydrodynamic conditions and INITIALLY LONGSHORE-UNIFORM cross-shore sandbar profile (as stated in L157-158: (alongshore...considered).

L142 z_b = mean bed level, not mean sea level

L246 fastest growing mode (instead of modes)?

L266 intermediate (without –s)

Figures 4 and 6: Why do the alongshore extents (y-axis limits) of the subplots vary? Wouldn't it be clearer (calmer) to make these the same?

Figure 4, middle row: For consistency, use "outer" bar pattern instead of "offshore"

Figure 7 labels x-axes and Figure 8 titles: for consistency, use small x (instead of X)

Figure 10 Mention somewhere that the colors refer to the modes in Figure 9.

---

## Author Comment (AC1) · 19 Mar 2020

We thank Reviewer 1 for the constructive comments. Our reply to each comment is shown below.

* Abstract: The sentence "For intermediate differences between the two sandbars depths, patterns on both longshore bars appear to be fully coupled" was quite unclear to me before going through the manuscript, and it should therefore be slightly rephrased as by "fully coupled" the author mean something more like the 2 bar patterns grow at a similar rate.

-> Done. It now reads: "For intermediate differences between the two sandbars depths, patterns on both longshore bars appear to be fully coupled (similar growth rates and

strongly correlated pattern shape)"

* Introduction: This is a very nice section providing background on coupling patterns. I think, however, that the authors should make clearer that nothing has been done on the influence of the distance between the bars, and crest depth difference (although for the latter it is tentatively said), in other words, the authors may put more emphasis onto what is new in their contribution. I was not comfortable with the use of the word 'geometry' in this section. To me the sandbar geometry refers to the barline (2D horizontal) and does not include depth, I would rather talk about 3D morphology, but I am not English native so my comment may not be relevant.

-> The comment is relevant. We have changed "geometry" for "3D morphology". We have kept "geometry" when dealing with the 2D cross-shore profile only.

* The authors should also refer to the work of Garnier et al (2013, GRL) when dealing with bar straightening under obliquely incident waves.

-> We added a reference to Garnier et al (2013) in the sentence were we discuss the role of wave obliquity in sandbar straightening.

* I also recommend to indicate that the authors will stick to shore-normally incident waves in the last paragraph of the introduction section.

-> The text has been modified to state, in the last paragraph of the Introduction, that the hydrodynamics is forced by shore-normally incident waves.

* Numerical Model: Please double check all notations: for instance L142 zb is not the mean sea level (zs in eq (1)) but the mean seabed elevation, h in eq(12) is not defined it should be the bathymetric perturbation (deviation from the basic state). It would be nice to add a short paragraph with the equation of the perturbation and indication of how \tau is computed (why referred to as growth time rather than e-folding time in most papers?). I understand that this is given in earlier papers, but that would help the reader to have a standalone paper.

Interactive
comment

[Figure]

-> Motivated by the comments of the reviewer, we have rewritten the description of the variables at the beginning of the section and revised the rest of the notations. At the end of the section, the linear stability analysis is now also described very briefly.

* Results: * I encourage to modify Fig. 4 to improve readability of arrows and perturbation with for each mode the left-hand panels water depth with iso-contours and right-hand panels perturbation h (not contoured) with currents. Please also indicate the time at which the different bathymetries have been plotted.

-> The figure has been modified in line with the recommendations of the reviewer. Since this is a linear stability analysis, it is not proper to refer to a time in which the different magnitudes are plotted but to the amplitude of the disturbances. To facilitate the visualization of the modes, the plots have been made for an arbitrary bottom perturbation of 0.5m. The size of the velocity vectors has also been adjusted to facilitate the visualization. The maximum velocity is indicated for each of the graphs. Figure 4 and the corresponding figure caption, which provides information on the amplitude of the bottom perturbation, have been modified in the manuscript.

** L198: "When the coupling between sandbars is obvious" I guess that the authors used some kind of more or less objective threshold in terms of ratio of perturbation amplitude at the inner and outer bar to discriminate between "obvious" and more "subtle" couplings, please clarify.

-> In the submitted manuscript we used our judgment to indicate if a sandbar was dominant or if the pattern was coupled. Stimulated by the reviewer we have looked at perturbation amplitudes and realized that our "judgment" is extremely similar to a more objective criteria. If the amplitude of the perturbation of one of the sandbars is over 80% larger than the amplitude of that in the other sandbar, we consider that only the sandbar with largest perturbation amplitude will develop into a crescentic sandbar. If the amplitude of either the inner and outer sandbars is between 40% and 80% larger, that sandbar will dominate the coupling. If the difference in the amplitude perturbation

is below 40%, the two sandbars are considered to be fully coupled. Similarly, if the amplitude of the perturbation close to the shoreline is at least 20% of the largest amplitude, we consider that also the shoreline is unstable. Using the above approach, results (and figures in the manuscript) change only slightly. We have adopted the new approach, and modified the text and changed the figures (results are essentially unchanged). The approach is presented in lines 203-209 of Section 3.

** L200: the same applies here, did the authors use a some kind of threshold in term of perturbation near the shoreline, e.g. at a given basic state iso-contour ?

-> See reply above

*Discussion: * Dealing with the limitations of the study is half of the discussion, I advises lightly to shorten the limitations, which should start as a new paragraph L270, and/or extend the first part of the discussion

-> We would really like to keep all the limitations we discuss as many readers might not be familiar with linear stability analysis.

* In the limitations part, the authors may add a couple of sentences on the fact that coupling at half of the outer bar wavelength (Castelle et al., 2010b) cannot be reproduced here.

-> This has been added in the Discussion. The new text reads: "Since the present model is linear, the concept of coupling is limited to the initial morphological formation and, since linear stability analysis focuses on the fastest growing wavelength, coupling at half of the outer bar wavelength cannot occur."

* My own 'empirical' knowledge of double barred beaches I've been to along different coasts is that out-of-phase coupling is much more common than in-phase coupling, this also applies to shorelines and bar coupling along single barred beaches. The results here indicate the couplings are about equally distributed between in-phase and out-of-phase. I do not necessarily ask the authors to discuss this, because my qualitative

observations may be biased and they may not think the same, but I am curious to know what the authors think about the coupling type predominance and potential mismatch with model outputs.

-> The observation from the reviewer is certainly relevant. Figure 7 shows that, for mode 1, when the outer bar dominates over the inner bar the coupling is out of phase. It is only when the inner bar dominates that we observe in-phase coupling. To draw a conclusion about the predominant configuration, many more cross-shore profiles should be studied so that results can be more easily generalized.

* Technical corrections: * The authors may consider adding 'linear stability analysis' and/or 'under shore-normally incident waves' in their title.

-> The title now reads "Emerging crescentic patterns in modeled double sandbar systems under normally-incident waves"

* The paper is written in very good English, however there are a few typos here and there (L116 missing bracket, remove comma at the end of equation (6), add '.' before

-> Done

* 'Following' L147, uncapitalize 'X' of the 3 top \Delta X in Fig. 8, idem in Fig. 7, ...) and recommend a very last proofread.

-> Figures 5, 7, 8 and 9 have been modified so that only Delta x (lower case) appears throughout the manuscript.

* Remove or increase label size in Fig. 6 (cross-shore/longshore distance, iso-contours)

-> The authors agree on the difficulty of reading both the labels and the arrows. For this reason only the isobaths are now shown in the figure.
* * *
[Figure]

2019.

---

## Author Comment (AC2) · 19 Mar 2020

We thank Reviewer 2 for the constructive comments. Our reply to each comment is shown below.

\* Besides elaborating on the motivation for the study and embedding of the results, it would be insightful for the reader if these authors in particular (given their shared experience with this type of model and other modelling approaches) could provide a (brief) reflection as to why LSA is particularly suitable for tackling this study.

-> This is now addressed in the introduction. The whole paragraph now reads "In this contribution we aim to systematically address the role of initial bathymetry on the coupling between sandbars, an area that so far has received only limited attention

\citep[see also ][]{price2014}. Specifically, we wish to investigate if sandbar coupling can freely emerge or if it is always the response of a sandbar to the development of a pattern in the other sandbar. We use linear stability analysis so that we can better focus on initial growth of the features and on the interactions that cause the emergence of the sandbar patterns. Adoption of a partly analytical approach also ensures the possibility of performing an exploration of the parameter space in a minimal amount of time, especially compared to nonlinear simulations. Other modelling studies of morphological evolution of double barred beaches also used linear stability analysis to analyze the depth- and wave-averaged equations coupled to sediment transport and morphological evolution. \cite{calvete07} used linear stability analysis to show that the initial cross-shore beach profile can be as important as wave height in determining the growth rate and alongshore spacing of crescentic bars. The work of \cite{klein06} for example showed that the magnitude of the longshore current and wave height are directly related to the preferred spacing and the growth rate, respectively."

* Would the use of, for example, a nonlinear model lead to similar conclusions regarding the emergence of the patterns? Why (not)? The discussion section includes a reflection on the use of LSA herein (L274-289), but please mention in the methodology section what makes LSA suitable for answering the research question.

-> We now discuss the suitability of LSA in the introduction. We also changed the discussion to address the different role of LSA and nonlinear models. The text now reads: "The transition from forced to fully coupled occurs smoothly in the parameter space that has been examined. Since our analysis of the model dynamics is linear, the concept of coupling is limited to the initial morphological formation and, since linear stability analysis focuses on the fastest growing wavelength, coupling at half of the outer bar wavelength cannot occur. Also, we do not simulate the nonlinear interactions between competing wavelengths, which might lead to coupling over longer time scales (days to weeks) or the final equilibrium configuration. Both important aspects can be studied using analysis that include nonlinear mode interactions and that are suited to

study the long-term evolution and possibly the equilibrium of these systems.. "

*OTHER COMMENTS * Abstract: Please include the aim, or knowledge gap, in the abstract. The abstract now starts with a description of the approach, followed by the description of the results in Line 5.

-> We have modified the opening of the abstract into: "Double sandbar systems often characterize the surfzone of wave-dominated beaches and display a variety of poorly-explained spatial configurations. Here, we explore the morphodynamic stability of double-barred beaches using a model based on linear stability analysis."

* L9 it is unclear what is meant by "inner bar-modes are dominant" -> Please describe what "inner-bar modes" are, and also what other modes there are.

-> We have avoided making reference to "inner-bar dominant modes". The text now reads: "Our analysis indicates that modes of which the amplitude of the inner sandbar perturbation is larger than that of the outer sandbar are dominant for large height/depth differences between the two sandbars crests and small offshore wave heights. Patterns related to the outer sandbar dominate for small values of the difference in sandbar depth."

* L92-96 Somewhere here, when introducing the use of the model, elaborate on the reason for opting for a model based on LSA.

-> Both reviewers have asked for this and we have changed the introduction and discussion accordingly.

* L177 Transverse bars are mentioned here for the first time -> Please mention these in the introduction section as features that may appear coupled to the sandbar pattern (Ribas et al 2014, Ocean Dynamics).

-> The introduction now reads: "The \cite{short93} model also indicates that beach configurations can involve coupling between the sandbars and/or coupling between the inner sandbar and the shoreline, where transverse sandbars can also be present

\citep{ribas2015}. Notice we prefer to refer to the paper by Ribas et al. (Review of Geophysics, 2015), in which a full review of mechanisms leading to transverse sandbars is given.

* L185-189 This paragraph belongs in the methods section, including figure 5. The choice for focusing on delta(D) and delta(x) should be elaborated upon, probably in the introduction ( somewhere in L78-92). Why not, for example, investigate the effect of changing the cross-shore slope (which, admittedly, inherently includes changes in cross-shore distance and bar depth, but also bar volume)? For sake of clarity, it is also worth noting that delta(D) here means changing the depth of the outer bar, while keeping the inner bar depth the same.

-> We have moved this paragraph to the end of the previous section, including the figure. We also specifically address in the text how we changed delta(D). We appreciate that the study could have been performed changing the beach slope but, since the beach slope changes with delta(x) and delta(D) we thought that a focus on those parameters would be more insightful.

* L287-288 Here it is mentioned that "model predictions are in qualitative agreement with observations of the Truc Vert double sandbar system". How do they agree? Please explain or show by means of a comparitive figure.

-> We have modified the text to clarify why they agree. We also noticed the text contained a wrong reference which we have now changed to Castelle et al. (2015). The new text reads: "Although the objective of this contribution is limited to a numerical analysis of the possible unstable patterns arising in double sandbar configurations, model predictions are in qualitative agreement with observations of the Truc Vert (France) double sand bar system (Castelle et al., 2015) where transverse bars are coupled to inner bars during moderate conditions, and inner-outer bar coupling is observed for more energetic conditions (we stress that parameter settings are not necessarily representative of Truc Vert)."

* L288-289 Here is mentioned that bathymetries of coupled sandbars are scarce, obstructing the comparison of the model with field observations. Could a general comparison of measured delta(D) give some insight into the validity or probability of the model results, or do you expect delta(D) to differ when bars couple? How do the findings of this study translate to future field studies? Please reflect on this.

-> Our study indicates that delta(D) is an important variable in determining the configuration of the emerging pattern. Bathymetric measurements before and after the emergence of sandbar patters would certainly allow model predictions.

The text now reads: "Lack of detailed and systematic measurements of bathymetric evolution of coupled sandbar systems remains the biggest obstacle to model testing in this area of research. We envisage that future development in the extraction of bathymetry from video images will be hugely beneficial to this area of research."

* L271-273 (and elsewhere) The model shows that large waves lead to a shoreline that couples to the outer bar. Does this correspond to the observation of coupling between shoreline embayments and the outer bar shape during a severe storm, by Castelle et al (2015)?

-> Although the objective of this contribution is limited to a numerical analysis of the possible emerging patterns arising in double sandbar configurations, model results are in qualitative agreement with observations of the Truc Vert (France) double sand bar system \citep{castelle2015}, where transverse bars are coupled to inner bars during moderate conditions, and inner-outer bar coupling is observed for more energetic conditions (we stress that parameter settings are not necessarily representative of Truc Vert). Lack of detailed and systematic measurements of bathymetric evolution of coupled sandbar systems remains the biggest obstacle to model testing in this area of research. We envisage that future development in the extraction of bathymetry from video images will be hugely beneficial to this area of research \citep{van2008beach}."

* L293-294 "Our results indicate .. single unstable mode." -> This is indeed a key point

of this study. This statement would be even stronger if it was posed as the problem or hypothesis you wish to tackle with this study (also see my comment above).

-> We have modified the Introduction where the aim of the study is discussed. It now reads: "In this contribution we aim to systematically address the role of initial bathymetry on the coupling between sandbars, an area that so far has received only limited attention (Price et al., 2014). Specifically, we wish to investigate if sandbar coupling can freely emerge or if it is only the response of a sandbar to the development of a pattern in the other sandbar."

*TECHNICAL CORRECTIONS * L9 two sandbars crests -> two sandbar crest

-> Done.

L84 THESE authors named this PHENOMENON

Done.

* L93 hydrodynamic conditions and INITIALLY LONGSHORE-UNIFORM cross-shore sandbar profile (as stated in L157-158: (alongshore...considered).

-> Done.

* L142 $z_b$ = mean bed level, not mean sea level

-> Done. Following comments by another reviewer, we have rewritten the description of the variables at the beginning of the section and revised the rest of the notations.

* L246 fastest growing mode (instead of modes)?

-> Done.

* L266 intermediate (without –s)

-> Done.

* Figures 4 and 6: Why do the alongshore extents (y-axis limits) of the subplots vary?

**ESurfD**
* * *
Interactive
comment

Wouldn't it be clearer (calmer) to make these the same?

-> The reason for the difference in extension along the coast is that always two wavelengths are shown. In the cross-shore direction, 700m is always displayed.

* Figure 4, middle row: For consistency, use "outer" bar pattern instead of "offshore"

-> We have modified text and figures and now only "outer" bar pattern" is used.

* Figure 7 labels x-axes and Figure 8 titles: for consistency, use small x (instead of X)

-> Figures 5, 7, 8 and 9 have been modified so that only Delta x (lower case) appears throughout the manuscript.

* Figure 10 Mention somewhere that the colors refer to the modes in Figure 9.

-> We have added "The colors refer to the modes in Figure 9" in the caption of Figure 10.